# Can Optic Nerve Sheath Images on a Thin-Slice Brain Computed Tomography Reconstruction Predict the Neurological Outcomes in Cardiac Arrest Survivors?

**DOI:** 10.3390/jcm11133677

**Published:** 2022-06-26

**Authors:** Sung Ho Kwon, Sang Hoon Oh, Jinhee Jang, Soo Hyun Kim, Kyu Nam Park, Chun Song Youn, Han Joon Kim, Jee Yong Lim, Hyo Joon Kim, Hyo Jin Bang

**Affiliations:** 1Department of Emergency Medicine, Seoul St. Mary’s Hospital, College of Medicine, The Catholic University of Korea, Seoul 06591, Korea; sunghojjang86@hanmail.net (S.H.K.); emsky@catholic.ac.kr (K.N.P.); ycs1005@catholic.ac.kr (C.S.Y.); hanjoon@catholic.ac.kr (H.J.K.); ny1117@catholic.ac.kr (J.Y.L.); liebestest@hanmail.net (H.J.K.); banghyojin87@gmail.com (H.J.B.); 2Department of Radiology, Seoul St. Mary’s Hospital, College of Medicine, The Catholic University of Korea, Seoul 06591, Korea; znee99@gmail.com; 3Department of Emergency Medicine, Eunpyeong St. Mary’s Hospital, College of Medicine, The Catholic University of Korea, Seoul 03312, Korea; unidgirl@catholic.ac.kr

**Keywords:** heart arrest, optic nerve sheath diameter, prognostication, neurological outcome

## Abstract

We analyzed the prognostic performance of optic nerve sheath diameter (ONSD) on thin-slice (0.6 mm) brain computed tomography (CT) reconstruction images as compared to routine-slice (4 mm) images. We conducted a retrospective analysis of brain CT images taken within 2 h after cardiac arrest. The maximal ONSD (mONSD) and optic nerve sheath area (ONSA) were measured on thin-slice images, and the routine ONSD (rONSD) and gray-to-white matter ratio (GWR) were measured on routine-slice images. We analyzed their area under the receiver operator characteristic curve (AUC) and the cutoff values for predicting a poor 6-month neurological outcome (a cerebral performance category score of 3–5). Of the 159 patients analyzed, 113 patients had a poor outcome. There was no significant difference in rONSD between the outcome groups (*p* = 0.116). Compared to rONSD, mONSD (AUC 0.62, 95% CI: 0.54–0.70) and the ONSA (AUC 0.63, 95% CI: 0.55–0.70) showed better prognostic performance and had higher sensitivities to determine a poor outcome (mONSD, 20.4% [95% CI, 13.4–29.0]; ONSA, 16.8% [95% CI, 10.4–25.0]; rONSD, 7.1% [95% CI, 3.1–13.5]), with specificity of 95.7% (95% CI, 85.2–99.5). A combined cutoff value obtained by both the mONSD and GWR improved the sensitivity (31.0% [95% CI, 22.6–40.4]) of determining a poor outcome, while maintaining a high specificity. In conclusion, rONSD was clinically irrelevant, but the mONSD had an increased sensitivity in cutoff having acceptable specificity. Combination of the mONSD and GWR had an improved prognostic performance in these patients.

## 1. Introduction

In the era of targeted temperature management (TTM), reliable prognostication of the neurological recovery in comatose cardiac arrest survivors has become more challenging [1]. Currently, the international guidelines for post-resuscitation care recommend multimodal prognostication composed of clinical examinations, neurophysiologic tests, serum biomarkers and brain imaging, including computed tomography (CT) scans and magnetic resonance imaging [2,3]. Among these, brain imaging provides a unique ability to visualize and quantify the structural brain injury [4]. According to a recent survey, the majority of intensivists recognized that a neurological examination alone is not enough to predict the outcome after cardiac arrest, and brain CT scan is considered one of most useful additional tests [5].

On CT, the amount of brain edema can be quantified as the gray-to-white matter ratio (GWR) [6,7,8,9,10,11,12], and currently, it has been suggested that a reduced GWR supports the presence of a poor prognosis in patients who remain comatose, especially when presented with other outcome predictors [2,3]. In addition, the optic nerve sheath diameter (ONSD) can be used as an outcome predictor [13,14,15,16]. The optic nerve is surrounded by cerebrospinal fluid (CSF), which is contiguous with intracranial CSF [15]. Increased intracranial pressure (ICP) is transmitted through this subarachnoid space causing distention of the dural optic nerve sheath, especially the retrobulbar segment [17], and the neurological outcomes can be predicted by measuring the ONSD. However, a recent meta-analysis showed that the mean ONSD of cardiac arrest survivors was approximately 5–7 mm [15], so it is unclear whether clinicians can accurately measure the ONSD on routine brain CT images with a slice thickness of approximately 5 mm. Recent advances in CT technology have enabled thin-slice brain imaging in most routine clinical CT scans [18]. These high-resolution CT images are useful to identify subtle skull base conditions and small structures such as the optic nerve [19]. Nonetheless, none of the previous studies have evaluated the ONSD using thin-slice images, and only a few studies have compared the ONSD to GWR [12,13,14].

In this study, we analyzed the prognostic performance of the ONSD on 0.6 mm thin-slice reconstruction images compared to being measured on 4 mm routine-slice images, and we evaluated whether ONSD measurements can provide additional information on predicting the neurological outcome.

## 2. Materials and Methods

### 2.1. Study Design and Subjects

This was a retrospective observational study of one tertiary hospital TTM registry taken between 2013 and 2018 in Seoul, Korea. We included adults aged ≥18 years who were treated with TTM after out-of-hospital cardiac arrest (OHCA), and whose brain CT imaging included thin-slice images. Since the ICP increases over time after cardiac arrest, to reduce heterogeneity, we excluded patients who had a brain CT scan more than 2 h after the return of spontaneous circulation (ROSC) from the study. Patients whose long-term outcome was not available or who had artifacts on brain CT were also excluded.

Our institutional ethics committee approved this study, and the requirement for consent was waived because of the retrospective nature of the study.

### 2.2. Brain CT Acquisitions and Post-Resuscitation Care

All patients routinely underwent nonenhanced brain CT scans immediately after ROSC. However, we could not examine brain CT in patients with hemodynamic instability or who had extracorporeal membrane oxygenation for hemodynamic support.

Sixty-four-channel scanners (Somatom Sensation 64; Siemens Medical Solutions, Erlangen, Germany) were used for all CT studies. The scanning parameters were as follows: 120 kVp, 380 mAs, FOV = 250 × 250 mm, matrix 512 × 512, and a slice thickness 0.6 mm. The clinical standard axial images were reconstructed with 4 mm-slice thickness and a standard kernel for soft tissue and a sharp kernel for the bone structures. In addition, the thin-slice (0.6 mm) axial images were reconstructed in the axial scans with a standard kernel.

After brain CT scanning, TTM at 33 °C or 36 °C for 24 h was started immediately. All comatose survivors after OHCA were treated in accordance with the local and international post-resuscitation care guidelines [20,21].

### 2.3. Interpretation of the Brain CTs

The interpretations of the brain CTs were retrospectively performed by investigators who were blinded to patient outcome. Two authors measured the ONSD and GWR together using a ruler and the regions of interest (ROI) function. From the 4 mm routine-slice images, GWRs and the routine ONSD (rONSD) were measured. GWR in the basal ganglia (GWR-BG) and in the centrum semiovale and the high convexity area (GWR-CB) were measured from the appropriate routine axial images (Appendix A) [10]. Finally, we calculated the average GWR (GWR-AV). The ONSD was measured according to the methods described in previous studies [13,22]. The optic nerve sheath image was magnified to 300% at a window width of 350 and a level of 40, and each ONSD was measured at a distance 3 mm behind the eyeball (Appendix A). Among these ONSD measurements, the highest value was selected, and the diameters that were measured for the left and right eyes were averaged to yield the rONSD value.

From the 0.6 mm thin-slice images, the maximal ONSD (mONSD) was measured using the same settings and method as previously mentioned (Appendix A). The optic nerve sheath area (ONSA) was calculated from all of the measured ONSDs on each axial image in which the optic nerve sheath was observed (Appendix A).

### 2.4. Outcome Measurement

The neurological outcome at 6 months after ROSC was evaluated via a face-to-face or telephone interview with the patients or relatives. We dichotomized the patients into good neurological outcome (Glasgow-Pittsburgh Cerebral Performance Category [CPC] score of 1 or 2) and poor neurological outcome (CPC score of 3–5) groups.

### 2.5. Statistical Analysis

The categorical variables are expressed as the number and percentage, and the continuous variables are expressed as the mean ± standard deviation or the median and interquartile range (IQR). The chi-square test, Student’s t test and the Mann-Whitney U test were used to compare the groups. To assess the performances of the parameters, the receiver operating characteristic (ROC) curve, the cutoff values and the sensitivities and specificities of the parameters, which were calculated using an exact binomial 95% confidence interval (CI), were evaluated. The intraclass correlation coefficient (ICC) and Pearson correlation coefficients between the predictors were calculated. We also created combined models with the variables using logistic regression models. Pairwise area under the ROC curve (AUC) comparisons were performed using the nonparametric approach [23].

All statistical analyses were performed using IBM SPSS version 24 software (IBM, Armonk, NY, USA). All *p* values were two-tailed, and *p* < 0.05 was considered significant.

## 3. Results

### 3.1. Characteristics of the Study Participants

Over the study period, 230 OHCA patients were treated with TTM. Among these, 55 patients did not undergo brain CT scans, and 16 patients were excluded from the analysis. Finally, a total of 159 patients were analyzed (Figure 1). After 6 months, 46 (28.9%) patients had a good neurological outcome, and 113 (71.1%) patients had a poor neurological outcome. Diabetes mellitus, non-shockable rhythm and noncardiac origin arrest were more common in the poor outcome group than in good outcome group (all *ps* < 0.05) (Table 1). The good outcome group were younger, and their arrest time was shorter than in the poor outcome group (both *ps* < 0.001). Immediately after ROSC, absent brainstem reflex, a Glasgow motor score < 3 and the absence of spontaneous respiration were more common in the poor outcome group (all *ps* < 0.001).

### 3.2. GWR Variables between the Outcome Groups

The median ROSC-to-CT intervals in the good and poor outcome groups were 24.5 (IQR, 14.8–33.5) min and 15.0 (IQR, 10.0–24.0) min, respectively. The mean GWR-BG and GWR-AV were significantly lower in the poor outcome group (1.24 ± 0.07 vs. 1.19 ± 0.07, *p* < 0.001; 1.21 ± 0.05 vs. 1.18 ± 0.06, *p* < 0.001, respectively), but there was no statistically significant difference in GWR-CB between the two groups (*p* = 0.068) (Table 1).

### 3.3. ONSD Variables between the Outcome Groups

There was no significant difference in rONSD between the outcome groups (6.22 ± 0.79 mm vs. 6.01 ± 0.75 mm, *p* = 0.116) (Table 1). The mean mONSD and the mean ONSA in the poor outcome group (7.22 ± 0.74 mm and 37.42 ± 6.93 mm^2^, respectively) were significantly higher than those in the good outcome group (6.92 ± 0.67 mm, *p* = 0.018; 34.20 ± 5.81 mm^2^, *p* = 0.006, respectively). Although there were strong positive correlations between rONSD and mONSD or ONSA (r = 0.852; r = 0.830, respectively) (Figure 2A,B), the ICC between rONSA and mONSD was 0.64 (95% CI, −0.15–0.88), and the mean mONSD (7.13 ± 0.73 mm) was increased by approximately 15% (0.97 ± 0.42 mm) compared with the mean rONSD (6.16 ± 0.78 mm). Figure 2C plots the difference between the two measurements (thin-slice diameter minus routine-slice diameter) against rONSD, which represented a moderately negative correlation (r = −0.374).

### 3.4. Prognostic Performances of the Single GWR or ONSD Variables Alone

GWR-AV (AUC 0.70 [95% CI, 0.62–0.77]) and GWR-BG (AUC 0.70 [95% CI, 0.62–0.77]) showed better prognostic performance in predicting a poor outcome than GWR-CB (AUC 0.63 [95% CI, 0.55–0.70]) (*p* = 0.030; *p* = 0.231, respectively) (Figure 3A). In the ONSD analyses, mONSD (AUC 0.62, 95% CI: 0.54–0.70) and ONSA (AUC 0.63, 95% CI: 0.55–0.70) were likely to have a better prognostic performance than rONSD (AUC 0.59, 95% CI: 0.51–0.67) although the differences were not significant (both *ps* > 0.05) (Figure 3B).

A GWR-AV value < 1.11 predicted a poor outcome with a sensitivity of 12.4% (95% CI, 6.9–19.9) and a specificity of 100% (95% CI, 92.3–100.0). In two good outcome patients, rONSD and mONSD were measured as 8.00 mm and 8.56 mm and as 7.64 mm and 8.90 mm, respectively (Appendix A). Finally, the cutoff values of each predictor that had a specificity of 100% (rONSD > 8.00 mm; mONSD > 8.90 mm; ONSA > 47.54 mm^2^) had very limited sensitivities (0.9%, 1.8%, 8.0%, respectively) (Table 2 and Figure 4). However, when selecting the cutoff values with a specificity of 95.7% (95% CI, 85.2–99.5), their sensitivities increased, especially on thin-slice imaging (7.1% [95% CI, 3.1–13.5], 20.4% [95% CI, 13.4–29.0] and 16.8% [95% CI, 10.4–25.0], respectively).

### 3.5. Association between the GWR and ONSD Variables and the Combined Models

The performance of a composite of GWR-AV and one of the ONSD variables was better for determining a poor outcome than the use of each ONSD variable alone, although there was no difference between these combination models (all *ps* > 0.05) (Figure 3C). Figure 4 represents the scatter plots showing the distribution of the GWR-AV and ONSD variables, and there was no correlation between these variables. Based on coordinate lines that had a 100% specificity for a poor outcome, only a small number of patients with poor outcomes were located in zones that were associated with lower GWR-AV or higher ONSD values. When we used GWR-AV values of <1.11 or ONSD variables that had cutoffs with a specificity of 95.7% (95% CI, 85.2–99.5), the sensitivities improved to 19.5% (95% CI, 12.6–28.0), 31.0% (95% CI, 22.6–40.4) and 27.4% (95% CI, 19.5–36.6), respectively, while maintaining a high specificity (Table 2).

Considering that the brain CTs were undertaken in the early phase after ROSC, we also investigated whether the prognostic performance was improved when combining the ONSD variables with early risk assessment models using the GWR-AV and resuscitation variables (non-shockable rhythm and arrest time) or when combined with the neurological variables (absent brainstem reflex and absent motor response). Finally, after adding any of the ONSD variables, none of the prognostic performances were significantly increased (Table 3).

## 4. Discussion

In this study, we analyzed the early brain CT scans within 2 h after ROSC and measured mONSD and the ONSA on thin-slice images to compare routine-slice images for outcome prediction. While there was no significant difference in rONSD between the outcome groups, the mONSD had an increased sensitivity in cutoff having acceptable specificity. In particular, the usefulness of thin-slice brain images was observed, especially when used in combination with GWR.

Increased ICP (IICP) is associated with poor neurological outcomes in comatose cardiac arrest patients [24,25]. Instead of a direct ICP measurement via an invasive procedure, the measurement of the ONSD on brain CT or ocular ultrasonography can be used as a useful indirect measurement method [26]. However, our results failed to show a statistical difference in rONSD between the outcome groups. We speculate that this was due to the limited reliability of the ONSD measurement on routine-slice brain CTs and the timing of measurements. The ONSD, as measured from a brain CT, may have more practical advantages than ultrasonography because it does not require the availability of additional experts. However, the clinical relevance has certainly been questioned by researchers [14,22]. In one pioneering study, the ONSD on routine-slice brain CT within 24 h after ROSC had excellent discriminative power for the outcome prediction [13]. However, the following results were inconsistent among the different studies [16,22]. A large-scale study that included 329 routine-slice brain CTs within 2 h was not able to show that there were any differences in the ONSD between the outcome groups [22]. In a recent meta-analysis, the ONSD on brain CT images showed a similar specificity to the ONSD on ultrasonography, but with significantly lower sensitivity [16]. According to one study that evaluated serial ultrasonography, an increase in the ONSD had the highest sensitivity at 24 h after ROSC because the patients’ IICP were reported beginning 24 h after the ROSC in a poor outcome group treated with TTM [27,28,29]. Accordingly, our timing of the CTs could have caused the results of this study to be different from the results of previous studies which included brain CTs with wider time intervals.

To the best of our knowledge, in cardiac arrest patients, the measurement of the ONSD on thin-slice images has not been studied previously. The mean mONSD was measured to be higher than the mean rONSD by approximately 1 mm (15%). The smaller the that the measurement of the rONSD was, the greater the difference between the two methods. This finding suggests that the routine-slice images could not actually reflect the true ONSD, although the measurement reliability could increase in patients with IICP. Therefore, we believe that the ONSD measurement using thin-slice images will lead to better prognostic performance. Since routine- and thin-slice images can be simultaneously reconstructed by raw data obtained by thin-slice thickness, there is no concern about any increased time and additional radiation exposure. Nonetheless, mONSD and ONSA still had limited sensitivities. Inflammatory processes of the optic nerve sheath (as in optic neuritis, multiple sclerosis and vasculitis) can cause perineural edema that increases the ONSD [30,31]. Additionally, since the ONSD can vary from person to person, the ONSD may not fully reflect the IICP [32]. Thus, using the ONSD measurement in isolation to predict a poor outcome is unwarranted, and it should only be used as one test within a multimodal prognostic approach.

Our findings regarding GWR were consistent with those published in previous literature. Recent systematic reviews showed that there were methodological heterogeneities [7,33,34]. Therefore, the prognostic performance of these variables varied widely among the different studies. Although Streitberger et al. suggested that a CT performed >24 h after ROSC is an important prognostic tool [35], the optimal timing of brain CT is still unknown. Generally, the CT scans to find the etiology of cardiac arrest are used to predict the neurological outcome in patients who are still comatose 3 days after ROSC [2,3], and according to recent studies with CT acquisition timing similar to our study, a GWR-AV predicted a poor outcome with relatively low sensitivity (3.5–20.3%) [9,10,14]. As the GWR decreases over time in patients with a severe hypoxic-ischemic brain injury, its discriminative performance and sensitivity increase over time [35,36,37,38]. Thus, a GWR that is determined within this time window may not be a good outcome predictor [6,39].

Interestingly, we also found discrepancies between the GWR and ONSD results in each patient. Accordingly, the combination of the GWR and ONSD results, especially on thin-slice CT images, improved the sensitivity (31.0%) with an acceptable specificity (95.7%) for a poor outcome although their CIs overlapped that of GWR or ONSD alone. which is in line with the results of previous literatures [13,14]. In the current era of TTM and multimodal prognostication, it is important to be aware that additional measurements other than GWR from brain CT images can provide additional prognostic information during the early phase, when other prognostic tests have not been evaluated. Moreover, the early stratification of brain injury patients can help clinicians optimize the doses of the medications used for in-hospital treatments [40]. Although we failed to find a significant improvement after adding the ONSD variables for an early risk assessment, further studies are warranted to confirm the usefulness of the ONSD measurement in different settings.

Several limitations should be carefully considered while interpreting our results. First, this study was retrospectively performed in a single hospital and included a relatively small number of patients with good outcomes. There may be selection bias because there were not enough patients who had a noticeable difference in their ONSDs between the two measurement methods included in this analysis. We found that a combination of the GWR and mONSD improved the sensitivity, but our study may be statistically underpowered. We could also not statistically adjust for the other variables, such as shock, that could cause an additional neurological injury after the brain CT imaging. Second, the measurement method using a ruler and ROI is often subjective. Although the measurement of the ONSD on brain CT is known to have high interrater reliability [13,14], in our analysis, two researchers interpreted the CTs together, and the interrater reliability was not calculated. Further research studies using automated and rater-independent methods are needed to establish the reliability of the measurements obtained on brain CT images for prognostication [41]. Third, our inclusion criteria had a strict time window. Therefore, we do not recommend using our 100% specificity cutoff in patients who underwent brain CTs using different measurement settings. Fourth, as with most prognostication studies, the brain CT results were not blinded, which could have influenced the decisions regarding withholding advanced treatment.

## 5. Conclusions

In this analysis, which included the brain CT scans within 2 h following ROSC, rONSD was not clinically relevant for outcome prediction. Although the ONSD variables on thin-slice brain CT had limited sensitivities to predict a poor neurological outcome, combination with GWR improved the sensitivity for determining poor neurological outcomes while maintaining a high specificity.

## Figures and Tables

**Figure 1 jcm-11-03677-f001:**
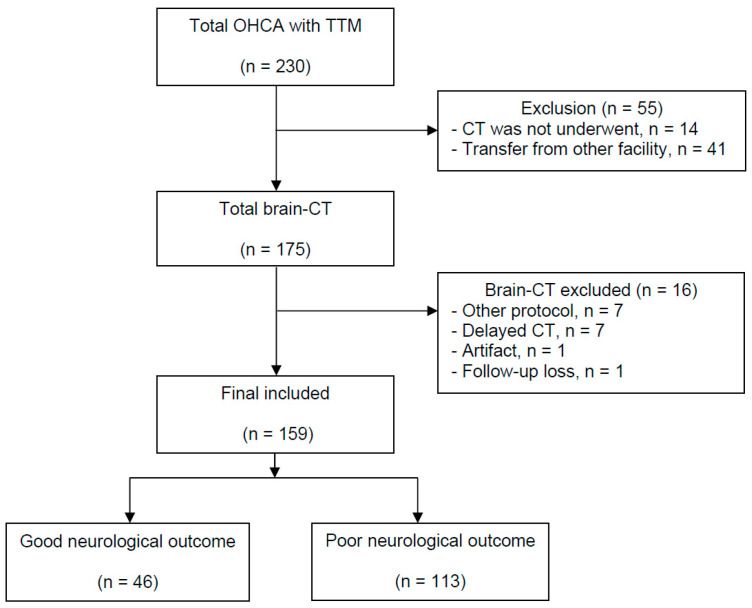
Flow diagram of the included patients. OHCA, out-of-hospital cardiac arrest; TTM, targeted temperature management; CT, computed tomography.

**Figure 2 jcm-11-03677-f002:**
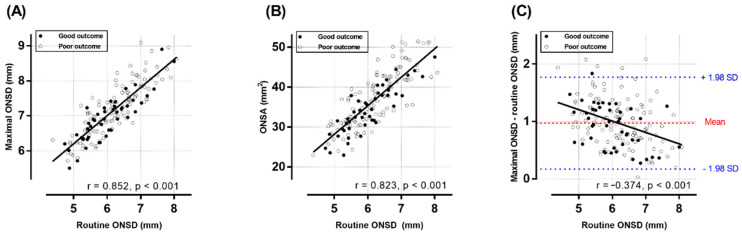
The correlation between the routine ONSD on 4 mm-slice CT images and the maximal ONSD or ONSA on 0.6 mm-slice CT images. The scatter plot and linear regression lines show the correlation between the routine ONSD and maximal ONSD (**A**) or the ONSA (**B**). The difference between the measurements (maximal ONSD minus routine ONSD) are plotted against the routine ONSD (**C**). If the differences between the 2 measurement techniques are small, the plot should center near zero. The red dashed line depicts the mean of the differences; the blue dotted lines denote the limits of agreement (mean ± 1.96 times of SD). The Pearson correlation coefficients (r) and *p* values are indicated. ONSD: optic nerve sheath diameter, ONSA: optic nerve sheath area, SD: standard deviation.

**Figure 3 jcm-11-03677-f003:**
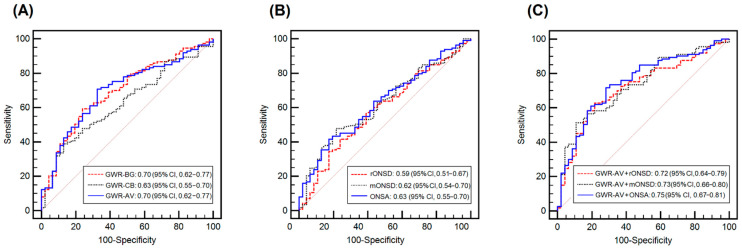
Receiver operating characteristic curve for the prediction of a poor 6-month neurologic outcome. (**A**) The AUCs for GWR-BG, GWR-CB and GWR-AV. (**B**) The AUCs for rONSD, mONSD and the ONSA. (**C**) The AUCs for the combined model with GWR-AV and rONSD, GWR-AV and mONSD, and GWR-AV and the ONSA. AUC: area under the curve, GWR-BG: gray-to-white matter ratio at the basal ganglia level, GWR-CB; gray-to-white matter ratio at the centrum semiovale and high convexity level, GWR-AV: average gray-to-white matter ratio, rONSD, routine optic nerve sheath diameter; mONSD, maximal optic nerve sheath diameter, ONSA, optic nerve sheath area.

**Figure 4 jcm-11-03677-f004:**
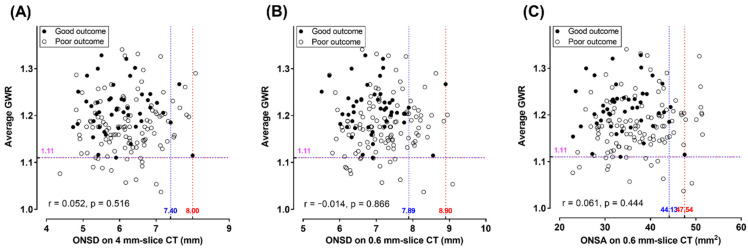
Scatter plots showing the distribution of the average gray-to-white matter ratio and optic nerve sheath diameters on 4 mm-slice CT (**A**) and 0.6 mm-slice CT images (**B**) or the optic nerve sheath area (**C**) in the good and poor neurological outcome groups. The Pearson correlation coefficients (r) and *p* values are indicated. The red dotted line indicates the cutoff value with a 100% specificity for determining a poor neurological outcome. The blue dotted line indicates the best cutoff value that was selected by the authors. GWR, gray-to-white matter ratio; ONSD, optic nerve sheath diameter; ONSA, optic nerve sheath area.

**Table 1 jcm-11-03677-t001:** Baseline characteristics of the included patients.

	Good Outcome (*n* = 46)	Poor Outcome (*n* = 113)	*p*-Value
Male	33 (71.7)	82 (72.6)	0.916
Age, years, mean ± SD	49.9 ± 16.1	61.1 ± 16.2	<0.001
Comorbidity			
Coronary artery disease	6 (13.0)	14 (12.4)	0.910
Hypertension	17 (37.0)	49 (43.4)	0.457
Diabetes mellitus	6 (13.0)	35 (31.0)	0.019
Chronic renal failure	3 (6.5)	8 (7.1)	0.900
Resuscitation variables			
Witnessed	32 (69.6)	77 (68.1)	0.861
Bystander CPR	34 (73.9)	70 (61.9)	0.150
Shockable rhythm	37 (80.4)	22 (19.5)	<0.001
Cardiac origin	44 (95.7)	59 (52.2)	<0.001
Time from arrest to ROSC, min (IQR)	15.0 (10.0–22.8)	34.0 (23.0–42.0)	<0.001
Variables immediately after ROSC			
Absent brainstem reflex	12 (26.1)	92 (81.4)	<0.001
Motor grade < 3	29 (63.0)	106 (93.8)	<0.001
Absent spontaneous respiration	19 (42.2)	92 (81.4)	<0.001
STEMI	12 (26.1)	19 (16.8)	0.181
STEMI or new onset LBBB	14 (30.4)	23 (20.4)	0.173
ROSC-to-CT interval, min (IQR)	24.5 (14.8–33.5)	15.0 (10.0–24.0)	0.003
GWR variables			
GWR-average	1.21 ± 0.05	1.18 ± 0.06	<0.001
GWR-basal ganglia	1.24 ± 0.07	1.19 ± 0.07	<0.001
GWR-cerebrum	1.19 ± 0.06	1.17 ± 0.08	0.068
ONSD variables			
Routine ONSD ^a^, mm	6.01 ± 0.75	6.22 ± 0.79	0.116
Maximal ONSD ^b^, mm	6.92 ± 0.67	7.22 ± 0.74	0.018
Difference of ONSDs, mm	0.91 ± 0.37	1.00 ± 0.43	0.224
Percentage difference, %	14.43 ± 6.24	15.17 ± 6.78	0.521
ONSA ^b^, mm^2^	34.20 ± 5.81	37.42 ± 6.93	0.006
Target temperature, 33 °C	42 (91.3)	103 (91.2)	0.975
Shock during the initiation of TTM	11 (23.9)	60 (53.1)	0.001

Data are presented as *n* (%) for the categorical variables unless otherwise indicated. ^a^ The routine ONSD was measured on 4 mm-slice brain CT images. ^b^ The maximal ONSD and ONSA were measured on 0.6 mm-slice brain CT images. SD, standard deviation; CPR, cardiopulmonary resuscitation; ROSC, return of spontaneous circulation; IQR, interquartile range; STEMI, ST segment elevation myocardial infarction; LBBB, left bundle branch block; CT, computed tomography; GWR, gray-to-white matter ratio; ONSD, optic nerve sheath diameter; ONSA, optic nerve sheath area, TTM, targeted temperature management.

**Table 2 jcm-11-03677-t002:** Prognostic accuracies of the predictors from early brain computed tomography for determining a poor 6-month neurological outcome.

	Cutoff	TP	FP	TN	FN	Sensitivity (95% CI)	Specificity (95% CI)
GWR-AV	<1.20 ^a^	80	15	31	33	70.8 (61.5–79.0)	67.4 (52.0–80.5)
	<1.11 ^b,^^c^	14	0	46	99	12.4 (6.9–19.9)	100.0 (92.3–100.0)
rONSD (mm)	>6.45 ^a^	47	11	35	66	41.6 (32.4–51.2)	76.1 (61.2–87.4)
	>7.40 ^b^	8	2	44	105	7.1 (3.1–13.5)	95.7 (85.2–99.5)
	>8.00 ^c^	1	0	46	112	0.9 (0.0–4.8)	100.0 (92.3–100.0)
mONSD (mm)	>7.28 ^a^	54	10	36	59	47.8 (38.3–57.4)	78.3 (63.5–89.1)
	>7.89 ^b^	23	2	44	90	20.4 (13.4–29.0)	95.7 (85.2–99.5)
	>8.90 ^c^	2	0	46	111	1.8 (0.2–6.3)	100.0 (92.3–100.0)
ONSA (mm^2^)	>39.21 ^a^	47	8	38	66	41.6 (32.4–51.2)	82.6 (68.6–92.2)
	>44.13 ^b^	19	2	44	94	16.8 (10.4–25.0)	95.7 (85.2–99.5)
	>47.54 ^c^	9	0	46	104	8.0 (3.7–14.6)	100.0 (92.3–100.0)
GWR-AV + rONSD (mm)	<1.11 ^b,^^c^ or >7.40 ^b^	22	2	44	91	19.5 (12.6–28.0)	95.7 (85.2–99.5)
	<1.11 ^b,^^c^ or >8.00 ^c^	15	0	46	98	13.3 (7.6–21.0)	100.0 (92.3–100.0)
GWR-AV + mONSD (mm)	<1.11 ^b,c^ or >7.89 ^b^	35	2	44	78	31.0 (22.6–40.4)	95.7 (85.2–99.5)
	<1.11 ^b,^^c^ or >8.90 ^c^	15	0	46	98	13.3 (7.6–21.0)	100.0 (92.3–100.0)
GWR-AV + ONSA (mm^2^)	<1.11 ^b,^^c^ or >44.13 ^b^	31	2	44	82	27.4 (19.5–36.6)	95.7 (85.2–99.5)
	<1.11 ^b,^^c^ or >47.54 ^c^	22	0	46	91	19.5 (12.6–28.0)	100.0 (92.3–100.0)

Data are presented as *n* for categorical variables unless otherwise indicated. ^a^ The cutoff value selected by the Youden index. ^b^ The cutoff value selected by the authors. ^c^ The cutoff value had a 100% specificity for determining a poor neurological outcome. TP, true positive; FP, false positive; FN, false negative; TN, true negative; CI, confidence interval; GWR-AV, average gray-to-white matter ratio; rONSD, routine optic nerve sheath diameter; mONSD, maximal optic nerve sheath diameter, ONSA, optic nerve sheath area.

**Table 3 jcm-11-03677-t003:** Areas under the receiver operating characteristic curves of the different models predicting 6-month poor neurological outcome.

	Crude ^a^	Model 1 ^b^	Model 2 ^c^	Model 3 ^d^	*p* ^a,b^	*p* ^a,c^	*p* ^a,d^
AUC of the resuscitation variable model (95% CI)	0.90 (0.83–0.96)	0.90 (0.83–0.96)	0.90 (0.84–0.96)	0.90 (0.84–0.96)	1.000	0.638	0.557
AUC of the neurological examination model (95% CI)	0.87 (0.82–0.93)	0.87 (0.82–0.93)	0.87 (0.82–0.93)	0.87 (0.82–0.93)	0.653	0.757	0.537

^a^ Resuscitation variable model: non-shockable rhythm + time from arrest to return of spontaneous circulation + GWR-AV. ^a^ Neurological examination model: absent brainstem reflex + a motor grade < 3 + GWR-AV. ^b^ Model 1: crude + routine ONSD on 4 mm-slice computed tomography. ^c^ Model 2: crude + maximal ONSD on 0.6 mm-slice computed tomography. ^d^ Model 3: crude + ONSA on 0.6 mm-slice computed tomography. AUC, area under the curve; GWR-AV, average gray–to-white matter ratio; ONSD, optic nerve sheath diameter; ONSA, optic nerve sheath area.

## Data Availability

The data presented in this study are available on request from the corresponding author. The data are not publicly available due to legal restrictions.

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
