# Peer review of "Can Optic Nerve Sheath Images on a Thin-Slice Brain Computed Tomography Reconstruction Predict the Neurological Outcomes in Cardiac Arrest Survivors?"

_jcm, 2022, doi:10.3390/jcm11133677_

Round 1

Reviewer 1 Report

Well described introduction.

The scientific background from which the study moves and the objectives of the study are clear.

Well designed retrospective study.

Precise description of the measurements made in accordance with the most appropriate scientific studies already published.

Adequate statistical analysis.

Critical discussion of the results, appropriately highlighting the innovation made.

Author Response

Response to Reviewer 1 Comments

Well described introduction.

The scientific background from which the study moves and the objectives of the study are clear.

Well designed retrospective study.

Precise description of the measurements made in accordance with the most appropriate scientific studies already published.

Adequate statistical analysis.

Critical discussion of the results, appropriately highlighting the innovation made.

We really appreciate you taking the time to review our manuscripts and positive comments.

Reviewer 2 Report

Summary

Kwon et al evaluated in a retrospective single centre study the prognostic accuracy for poor 6-month neurological outcome after ROSC of optic nerve sheath diameter and area on thin-17 slice (0.6 mm) brain computed tomography reconstruction images as compared to routine-slice (4 mm) images. I found this article of clinical interest. However, some minor issues should be addressed.

Results

In inter-test comparisons, the comparison of diagnostic accuracy should be made by comparing AUC of ROC with De Long's method. Furthermore, the comparison of the different sensitivities is of minor importance in the absence of a statistically significant difference between the AUCs of ROCs. I think this should be emphasised in the results and limitations.

Author Response

Response to Reviewer 2 Comments

Summary

Kwon et al evaluated in a retrospective single centre study the prognostic accuracy for poor 6-month neurological outcome after ROSC of optic nerve sheath diameter and area on thin-17 slice (0.6 mm) brain computed tomography reconstruction images as compared to routine-slice (4 mm) images. I found this article of clinical interest. However, some minor issues should be addressed.

à We really appreciate you taking the time to review our manuscripts and excellent critique and helpful suggestions for our manuscript. We have carefully reviewed your comments, responded to the comments and revised our manuscript accordingly. We believe that it has enhanced the quality of the manuscript.

Results

In inter-test comparisons, the comparison of diagnostic accuracy should be made by comparing AUC of ROC with De Long's method. Furthermore, the comparison of the different sensitivities is of minor importance in the absence of a statistically significant difference between the AUCs of ROCs. I think this should be emphasised in the results and limitations.

We totally agree with your concern. In the revised manuscript, we have added the p-values of DeLong's method and a sentence regarding statistical limitations as follows.

<Results section>

GWR-AV (AUC 0.70 [95% CI, 0.62–0.77]) and GWR-BG (AUC 0.70 [95% CI, 0.62–0.77]) showed better prognostic performance in predicting a poor outcome than GWR-CB (AUC 0.63 [95% CI, 0.55–0.70]) (p = 0.030; p = 0.231, respectively) (Figure 3A). In the ONSD analyses, mONSD (AUC 0.62, 95% CI: 0.54–0.70) and ONSA (AUC 0.63, 95% CI: 0.55–0.70) were likely to have a better prognostic performance than rONSD (AUC 0.59, 95% CI: 0.51–0.67) although the differences were not significant (both ps > 0.05) (Figure 3B).

<Discussion section>

Interestingly, we also found discrepancies between the GWR and ONSD results in each patient. Accordingly, the combination of the GWR and ONSD results, especially on thin-slice CT images, improved the sensitivity (31.0%) with an acceptable specificity (95.7%) for a poor outcome although their CIs overlapped that of GWR or ONSD alone. This finding is in line with the results of previous literatures [13,14].

<limitation paragraph in discussion section>

First, this study was retrospectively performed in a single hospital and included a relatively small number of patients with good outcomes. There may be selection bias because there were not enough patients who had a noticeable difference in their ONSDs between the two measurement methods included in this analysis. We found that a combination of the GWR and mONSD improved the sensitivity, but our study may be statistically underpowered. We could also not statistically adjust for the other variables, such as shock, that could cause an additional neurological injury after the brain CT imaging.

Reviewer 3 Report

In the manuscript titled “Can optic nerve sheath images on a thin-slice brain computed tomography reconstruction predict the neurological outcomes in cardiac arrest survivors?” the authors have analyzed the brain CT scans within two hours after the return of spontaneous circulation (ROSC) and measured maximal optic nerve sheath diameter (mONSD) and optic nerve sheath area (ONSA) on thin slices to gain information for outcome prediction. From this study, the authors conclude that both the parameters were not clinically relevant for outcome prediction. However, the ONSD variable on thin slices when combined with GWR improved the sensitivity of determining the outcome.

General comments:

The authors have done a nice study with the measurement of ONSD on thin-slice CT images in cardiac arrest patients. The authors have done a thorough statistical analysis and have provided a very good discussion for their study including the variability between the patients and the limitations of the study. I have the following comments:

Major comments:

1. On what basis did the authors choose to use the 2hr time point for their study? Authors should include an explanation for this in the manuscript.

This is important because a similar 2-hour study on routine slice brain (but not on thin slices) showed no difference in the ONSD (Ref 22, cited by the authors).

2. The authors provide contradictory statements regarding the use of ONSD to predict the outcome. In lines 289-291 the authors state that “Thus, using the ONSD measurement in isolation to predict a poor outcome is unwarranted, and it should only be used as one test within a multimodal prognostic approach.” However, in the conclusion (lines 335-338) the authors state that “On the other hand, the ONSD variables on thin-slice brain CT were likely to have a better prognostic performance and combining them with GWR improved the sensitivity for determining poor neurological outcomes while maintaining a high specificity.”

The authors should correct their statements in the conclusion by removing the sentence “the ONSD variables on thin-slice brain CT were likely to have a better prognostic performance”  and including “that ONSD when combined with GWR, improved the sensitivity for determining poor neurological outcomes.”

Also, in the conclusion, the authors should answer the title question “Can optic nerve sheath images on a thin-slice brain computed tomography reconstruction predict the neurological outcomes in cardiac arrest survivors?”

Minor comments:

1. The authors should increase the size of the main figures as they are very small and hard to read.

2. Authors have provided supplementary figures but have not mentioned them anywhere in the main text. The authors should cite them at the appropriate locations in the manuscript.

Author Response

Response to Reviewer 3 Comments

In the manuscript titled “Can optic nerve sheath images on a thin-slice brain computed tomography reconstruction predict the neurological outcomes in cardiac arrest survivors?” the authors have analyzed the brain CT scans within two hours after the return of spontaneous circulation (ROSC) and measured maximal optic nerve sheath diameter (mONSD) and optic nerve sheath area (ONSA) on thin slices to gain information for outcome prediction. From this study, the authors conclude that both the parameters were not clinically relevant for outcome prediction. However, the ONSD variable on thin slices when combined with GWR improved the sensitivity of determining the outcome.

General comments:

The authors have done a nice study with the measurement of ONSD on thin-slice CT images in cardiac arrest patients. The authors have done a thorough statistical analysis and have provided a very good discussion for their study including the variability between the patients and the limitations of the study. I have the following comments:

Thank you for allowing us to resubmit our paper and for your insightful comments. We have carefully reviewed your comments and revised the content of the manuscript as suggested. We strongly believe that your recommendations have enhanced the quality of our manuscript.

Major comments:

  1. On what basis did the authors choose to use the 2hr time point for their study? Authors should include an explanation for this in the manuscript.

This is important because a similar 2-hour study on routine slice brain (but not on thin slices) showed no difference in the ONSD (Ref 22, cited by the authors).

We totally agree with your concern. Generally, patients routinely underwent nonenhanced brain CT scans immediately after ROSC to identify brain hemorrhage and to decide whether TTM is indicated. Furthermore, because the ICP increases over time after cardiac arrest, to reduce heterogeneity, we excluded patients who had a brain CT scan more than 2 h after the ROSC from the study. In revised manuscript, we modified the methods section as follows.

Because the ICP increases over time after cardiac arrest, to reduce heterogeneity, we excluded patients who had a brain CT scan more than 2 h after the return of spontaneous circulation (ROSC) from the study. Patients whose long-term outcome was not available or who had artifacts on brain CT were also excluded.

  1. The authors provide contradictory statements regarding the use of ONSD to predict the outcome. In lines 289-291 the authors state that “Thus, using the ONSD measurement in isolation to predict a poor outcome is unwarranted, and it should only be used as one test within a multimodal prognostic approach.” However, in the conclusion (lines 335-338) the authors state that “On the other hand, the ONSD variables on thin-slice brain CT were likely to have a better prognostic performance and combining them with GWR improved the sensitivity for determining poor neurological outcomes while maintaining a high specificity.”

The authors should correct their statements in the conclusion by removing the sentence “the ONSD variables on thin-slice brain CT were likely to have a better prognostic performance” and including “that ONSD when combined with GWR, improved the sensitivity for determining poor neurological outcomes.”

Also, in the conclusion, the authors should answer the title question “Can optic nerve sheath images on a thin-slice brain computed tomography reconstruction predict the neurological outcomes in cardiac arrest survivors?”

As your recommendation, we removed the sentence and answered the title question in the conclusion as follows.

In this analysis, which included the brain CT scans within 2 h following ROSC, rONSD was not clinically relevant for outcome prediction. Although, the ONSD variables on thin-slice brain CT had limited sensitivities to predict a poor neurological outcome, combination with GWR improved the sensitivity for determining poor neurological outcomes while maintaining a high specificity.

Minor comments:

  1. The authors should increase the size of the main figures as they are very small and hard to read.

In the revised manuscript, we have inserted high resolution figures.

  1. Authors have provided supplementary figures but have not mentioned them anywhere in the main text. The authors should cite them at the appropriate locations in the manuscript.

We have confirmed that all supplementary figures were cited in the main text.